# Cyclosporin A Increases Mitochondrial Buffering of Calcium: An Additional Mechanism in Delaying Mitochondrial Permeability Transition Pore Opening

**DOI:** 10.3390/cells8091052

**Published:** 2019-09-07

**Authors:** Jyotsna Mishra, Ariea J. Davani, Gayathri K. Natarajan, Wai-Meng Kwok, David F. Stowe, Amadou K.S. Camara

**Affiliations:** 1Department of Anesthesiology, Medical College of Wisconsin, Milwaukee, WI 53226, USA (J.M.) (A.J.D.) (G.K.N.) (W.-M.K.) (D.F.S.); 2Cardiovascular Center, Medical College of Wisconsin, Milwaukee, WI 53226, USA; 3Department of Pharmacology and Toxicology, Medical College of Wisconsin, Milwaukee, WI 53226, USA; 4Cancer Center, Medical College of Wisconsin, Milwaukee, WI 53226, USA; 5Research Service, Zablocki VA Medical Center, Milwaukee, WI 53295, USA; 6Department of Physiology, Medical College of Wisconsin, Milwaukee, WI 53226, USA

**Keywords:** cyclosporin A, mitochondria calcium buffering, mitochondria bioenergetics, mitochondria permeability transition pore, inorganic phosphate

## Abstract

Regulation of mitochondrial free Ca^2+^ is critically important for cellular homeostasis. An increase in mitochondrial matrix free Ca^2+^ concentration ([Ca^2+^]_m_) predisposes mitochondria to opening of the permeability transition pore (mPTP). Opening of the pore can be delayed by cyclosporin A (CsA), possibly by inhibiting cyclophilin D (Cyp D), a key regulator of mPTP. Here, we report on a novel mechanism by which CsA delays mPTP opening by enhanced sequestration of matrix free Ca^2+^. Cardiac-isolated mitochondria were challenged with repetitive CaCl_2_ boluses under Na^+^-free buffer conditions with and without CsA. CsA significantly delayed mPTP opening primarily by promoting matrix Ca^2+^ sequestration, leading to sustained basal [Ca^2+^]_m_ levels for an extended period. The preservation of basal [Ca^2+^]_m_ during the CaCl_2_ pulse challenge was associated with normalized NADH, matrix pH (pH_m_), and mitochondrial membrane potential (ΔΨ_m_). Notably, we found that in PO_4_^3−^ (P_i_)-free buffer condition, the CsA-mediated buffering of [Ca^2+^]_m_ was abrogated, and mitochondrial bioenergetics variables were concurrently compromised. In the presence of CsA, addition of P_i_ just before pore opening in the P_i_-depleted condition reinstated the Ca^2+^ buffering system and rescued mitochondria from mPTP opening. This study shows that CsA promotes P_i_-dependent mitochondrial Ca^2+^ sequestration to delay mPTP opening and, concomitantly, maintains mitochondrial function.

## 1. Introduction

Regulation of intra-mitochondrial free calcium ([Ca^2+^]_m_) is critical in cardiac physiology and pathophysiology. Under physiological conditions, a moderate increase in [Ca^2+^]_m_ is believed to stimulate key enzymes of the Krebs cycle and oxidative phosphorylation and to drive mitochondrial ATP production to match cellular energy demand [1,2]. In contrast, a pathological increase in [Ca^2+^]_m_ causes opening of the mitochondrial permeability transition pore (mPTP), a key factor in initiation of cell death [3,4]. Pathophysiological dysregulation of [Ca^2+^]_m_ is a primary mediator in cardiac ischemia and reperfusion (IR) injury, as Ca^2+^ overloading can lead to apoptosis [5,6,7].

[Ca^2+^]_m_ is regulated by a dynamic balance between mitochondrial Ca^2+^ uptake, intra-mitochondrial Ca^2+^ buffering, and mitochondrial Ca^2+^ release. Mitochondrial Ca^2+^ uptake is mediated primarily through the mitochondrial Ca^2+^ uniporter (MCU) [8,9,10], and is controlled by the large membrane potential (ΔΨ_m_: −180 to −200 mV) across the inner mitochondrial membrane (IMM). The ΔΨ_m_ in turn is generated by the flow of electrons and proton pumping along the respiratory chain complexes [11]. When [Ca^2+^]_m_ increases, this depolarizes ΔΨ_m_, which is compensated by enhanced H^+^ pumping/extrusion to alkalinize the matrix. Therefore, powerful, dynamic buffering of matrix pH (pH_m_) and Ca^2+^ are required to enable sufficient recovery of ΔΨ_m_ and to avoid overloading the matrix with a high [Ca^2+^]. Inorganic phosphate (P_i_) has been recognized as a major player in maintaining the trans-matrix pH gradient when accompanied by the effective cotransport of H^+^ [12] and buffering of matrix Ca^2+^ through the formation of amorphous calcium phosphate (Ca–P_i_) granules [13,14,15]. The Ca–P_i_ buffer system sets the free Ca^2+^ at a steady-state level, enabling greater mitochondrial Ca^2+^ loading without impeding the Ca^2+^ uptake and affecting the efflux system [16,17,18]. The efflux systems that regulate [Ca^2+^]_m_ are the Na^+^/Ca^2+^exchanger (NCLX) [17], and the putative Na^+^-independent Ca^2+^ exchanger/Ca^2+^-hydrogen exchanger (CHE) [19]. Any disruption in the uptake, and or impairment in the buffering or efflux of Ca^2+^ would disrupt the delicate balance of the [Ca^2+^]_m_ and lead to impaired bioenergetics and to opening of the mPTP [3,4].

The opening of the high conductance mPTP channel is associated with a high degree of mitochondrial swelling, dissipation of ΔΨ_m_, uncoupling of oxidative phosphorylation, membrane rupture and release of sequestered Ca^2+^, metabolites, and apoptotic signaling molecules [20,21,22,23]. Although the molecular components of the mPTP and its regulation remain largely unclear, cyclophilin D (Cyp D) is the only unambiguously recognized regulatory component of the mPTP. Cyp D is a mitochondrial matrix peptidyl-prolyl cis-trans isomerase (PPIase) that is translocated to the IMM during high matrix Ca^2+^ conditions; Cyp D is proposed to facilitate conformational changes in the putative mPTP core proteins thereby regulating pore opening [24,25,26].

Adenine nucleotides (AdN: ATP and ADP) have been implicated in the inhibition of Ca^2+^-dependent mPTP opening [27,28]. A previous study from our laboratory suggested that matrix AdN modulate [Ca^2+^]_m,_ potentially by increased buffering of [Ca^2+^]_m_ [29]. Oligomycin (OMN), an F_0_F_1_-ATP synthase inhibitor, influences the AdN (ATP/ADP) pool, and has been shown to modulate mPTP opening [30]. Cyclosporin A (CsA), a potent mPTP inhibitor is also believed to suppress pore opening by inhibiting matrix Cyp D, thereby preventing the Cyp D-induced conformational changes in mPTP core proteins [31,32]. CsA has long been known to desensitize mPTP from early opening during Ca^2+^ challenges by impeding Ca^2+^ interaction with Cyp D; however, the direct effects of CsA on the [Ca^2+^]_m_ buffering system have not been investigated systematically. It is worth noting that in a previous study from Chalmers and Nicholls [14], it was proposed that CsA enhances the Ca^2+^ loading capacity of mitochondria without changing the relationship between free [Ca^2+^]_m_ and total [Ca^2+^]_m_ during continuous Ca^2+^ infusion in isolated rat liver and brain mitochondria. Altschuld et al. [33] proposed that CsA increases mitochondrial Ca^2+^ influx and reduces its efflux. Later, Wei et al. [34] demonstrated that although CsA had no effect on MCU activity, it inhibited NCLX activity at higher concentrations. Altogether, these findings raise important questions about how CsA delays Ca^2+^-induced mPTP opening while increasing net [Ca^2+^]_m_ accumulation. Our study sought to answer these questions by (i) examining the effect of CsA during repeated CaCl_2_ challenges over an extended time-period on mitochondrial Ca^2+^ buffering, and (ii) by examining the underlying changes in bioenergetics during excessive Ca^2+^ overload.

To address our objective, we investigated systematically the effect of CsA on mitochondrial Ca^2+^ buffering and compared its effect with a known matrix buffering component, the AdN pool (OMN+ADP), by monitoring [Ca^2+^]_e_, [Ca^2+^]_m_, and key mitochondrial bioenergetics variables, ΔΨ_m_, pH_m_, and NADH (redox state), under conditions of repeated Ca^2+^ loading. Furthermore, we determined the effect of CsA on the rescue of buffering capability and bioenergetics of failing mitochondria just before mPTP opening. We found that CsA enhanced the sequestration of mitochondrial Ca^2+^, maintained [Ca^2+^]_m_ at a steady-state level, and markedly delayed mPTP opening. In addition, CsA preserved ΔΨ_m_, NADH, and pH_m_ during CaCl_2_ bolus challenges. However, in the absence of P_i_, this CsA-induced matrix Ca^2+^ sequestration was abrogated, and in turn led to the early mPTP opening. The results described herein reveal a novel way by which CsA modulates matrix Ca^2+^ sequestration to maintain [Ca^2+^]_m_, despite increased Ca^2+^ loading. CsA-mediated Ca^2+^ sequestration is likely achieved via a P_i_-dependent [Ca^2+^]_m_ buffering system that delays Ca^2+^-induced mPTP opening.

## 2. Materials and Methods

### 2.1. Materials

All chemical reagents were purchased from Sigma-Aldrich (St. Louis, MO, USA), unless stated otherwise. Fluorescent probes Fura-4F, Fura-4F^AM^, tetramethylrhodamine methyl ester perchlorate (TMRM) and 2′,7′-Bis-(2-Carboxyethyl)-5-(and-6)-carboxyfluorescein, acetoxymethyl ester (BCECF^AM^) were purchased from Life Technologies (Eugene, OR, USA).

### 2.2. Animals

Albino Hartley guinea pigs of both sexes weighing between 250 to 350 g were procured from Kuiper Rabbit Farm (Gary, IN, USA). All procedures were carried out in accordance with the National Institutes of Health (NIH) Guide for the Care and Use of Laboratory Animals (NIH Publication No. 85-23, revised 1996) and were approved by the Institutional Animal Care and Use Committee of the Medical College of Wisconsin.

### 2.3. Mitochondria Isolation

Mitochondria were isolated from guinea pig hearts as described previously [29,35,36]. Briefly, the guinea pig was anesthetized with an intraperitoneal injection of 30 mg ketamine plus 700 units of heparin, for anticoagulation, and the heart was rapidly excised and minced in ice-cold isolation buffer containing 200 mM mannitol, 50 mM sucrose, 5 mM KH_2_PO_4_, 5 mM MOPS, 1 mM EGTA, and 0.1% bovine serum albumin (BSA) at pH 7.15 (adjusted with KOH). The suspension was homogenized at low speed for 20 s in ice-cold isolation buffer containing 5 U/mL protease (from *Bacillus licheniformis*) and the homogenate was centrifuged at 8000× *g* for 10 min. The supernatant was discarded, and the pellet was suspended in 25 mL isolation buffer, and centrifuged at 850× *g* for 10 min. The supernatant was centrifuged further at 8000× *g* to yield the final mitochondrial pellet, which was suspended in isolation buffer and kept on ice until experimentation. All isolation procedures were performed at 4 °C and all experiments were conducted at room temperature. Protein concentration was determined by the Bradford method and the final mitochondrial suspension was adjusted to 12.5 mg protein/mL with isolation buffer.

The functional integrity of mitochondria was determined by the respiratory control index (RCI) as described before [29,37]. Mitochondria were energized with pyruvic acid (PA, 0.5 mM; pH 7.15, adjusted with KOH) followed by ADP (250 µM) addition. RCI was defined as the ratio of state 3 (after added ADP) to state 4 respiration (after complete phosphorylation of the added ADP). Only mitochondrial preparations with RCIs ≥ 10 were used to conduct further experiments.

### 2.4. Experimental Groups and Protocols

Two protocols (Protocol A and Protocol B) were used to assess the effect of CsA and AdN on mitochondrial Ca^2+^ handling and bioenergetics in normal and Ca^2+^-overloaded mitochondria, as shown in Figure 1. Protocol A investigated the ability of CsA and AdN to modulate mitochondrial Ca^2+^ handling and delay mPTP opening. To further substantiate CsA-mediated buffering of matrix Ca^2+^, Protocol B was designed to test the effectiveness of CsA and AdN on rescuing a failing mitochondrial Ca^2+^ buffering system from imminent mPTP opening. There were five experimental groups: vehicle (DMSO), CsA, ADP, OMN, and OMN+ADP. Experiments were also conducted in the presence of deionized H_2_O as another vehicle (not shown). Each group was subjected to two different experimental protocols (Protocol A and Protocol B) that differed in the order of treatment and addition of CaCl_2_ boluses to the mitochondrial suspension in experimental buffer.

Delayed opening of mPTP (Protocol A): At t = 0 s, the experiment was initiated by suspending 0.5 mg of isolated mitochondria into the experimental buffer containing 130 mM KCl, 5 mM K_2_HPO_4_, 20 mM MOPS, 1 mM EGTA, 0.1% BSA, and EGTA ~0.036–0.040 µM at pH 7.15 (adjusted with KOH). At t = 30 s, mitochondria were treated with DMSO (1 µM), ADP (250 µM), OMN (10 µM), OMN+ADP or CsA (0.5 µM); at t = 60 s, mitochondria were energized with PA (0.5 mM). At t = 180 s, CaCl_2_ bolus (20 μM final concentration) was added and subsequent CaCl_2_ boluses added at 5 min intervals until pore opening (Figure 1A). Note that all experiments were conducted under state 2 conditions, except in the ADP-and OMN+ADP-treated groups.

Rescue of mitochondria from mPTP opening (Protocol B): The mitochondrial suspension was exposed to repetitive boluses of CaCl_2_ (20 μM) as described in Protocol A; rescue of mitochondria from mPTP opening with the different treatments was carried out at 1 min of the last CaCl_2_ bolus in which mitochondria Ca^2+^ uptake was observed before pore opening (Figure 1B). The onset of mPTP opening was predicted based on calcium retention capacity (CRC) of the DMSO (control)-treated group for each day’s experiment. The pulse preceding mPTP opening observed in the control was the pulse chosen for targeted intervention in all subsequent experiments. In all experiments, extrusion of Ca^2+^ via the Na^+^/Ca^2+^ exchanger (NCLX) was prevented by conducting all the experiments in Na^+^-free conditions. That is, the respiration buffer, mitochondrial substrates, and all reagents/drugs were Na^+^-free to prevent activation of the NCLX. Some experiments were conducted in the presence of 10 μM CGP 37157 (Tocris Bioscience), an NCLX inhibitor, which ascertained there was no potential Na^+^ contamination in the respiration buffer from other sources [35,38,39].

### 2.5. Mitochondrial Function Measurements

Fluorescence spectrophotometry (Qm-8, Photon Technology International, Horiba, Birmingham, NJ, USA) was used to measure mitochondrial function, including mitochondria extra- and intra-matrix free [Ca^2+^] ([Ca^2+^]_e_ and [Ca^2+^]_m_, respectively), ΔΨ_m_, redox state (NADH), and pH_m_. Fura-4F penta-potassium salt (1 µM, Invitrogen™, Eugene, OR) was used to measure [Ca^2+^]_e_. For [Ca^2+^]_m_ measurements, mitochondria were incubated with Fura-4F AM (5 µM, Invitrogen™, Eugene, OR) for 30 min at room temperature (25 °C) followed by a final spin and resuspension to remove any residual dye. ΔΨ_m_ was assessed using the cationic lipophilic dye TMRM (1 µM, Invitrogen™, Eugene, OR, USA) in a ratiometric excitation approach [40]. NADH was measured by tissue autofluorescence, and matrix pH (pH_m_) was assessed by incubating mitochondria in 5 μM BCECF^AM^ (Invitrogen, Carlsbad, CA, USA) for 30 min at room temperature (25 °C) followed by a final spin and resuspension [29,35,38,39].

### 2.6. Measurements of Free Ca^2+^

Quantification of [Ca^2+^]_e_ and [Ca^2+^]_m_ were made using the fluorescent Ca^2+^ indicator probe Fura-4F with dual-excitation wavelengths (λ_ex_) at 340/380 nm and a single emission wavelength (λ_em_) at 510 nm. Ca^2+^ fluorescent intensities with Fura-4F are not influenced by background noise (e.g., NADH autofluorescence), so a background subtraction was unnecessary [38]. Fura-4F fluorescence ratios (F_340_/F_380_) were used to calculate [Ca^2+^] using the equation described by Grynkiewicz: [41].

(1)$$\left[{\mathrm{Ca}}^{2+}\right]=K_{d}\frac{S_{f2}}{S_{b2}}\frac{\left(R-R_{\min}\right)}{\left(R_{\max}-R\right)}.$$

The K_d_ value for Fura-4F binding to Ca^2+^ is 890 nM, which was described by us previously [38]. R is the ratio of the fluorescence intensities at λ_ex_ 340 and 380 nm, S_f2_/S_b2_ is the ratio of fluorescence intensities measured at λ_ex_ 380 nm in Ca^2+^-free (f)/Ca^2+^-saturated (Ca^2+^-bound, b) conditions. R_min_ (Ca^2+^-free) and R_max_ (Ca^2+^-saturated) are R values for Fura 4F, carried out after mPTP opening, adding 1 mM CaCl_2_, followed by 10 mM EGTA, pH 7.1. The free [Ca^2+^] in the buffer was calculated using an online version of MaxChelator program (http://www.stanford.edu/~cpatton/maxc.html) and accordingly, a standard curve was generated for the Fura-4F signal to the free [Ca^2+^] in the experimental solution by fitting to the Grynkiewicz equation, as described above in Equation 1 [41].

### 2.7. Calculation of Mitochondrial Ca^2+^ Buffering Capacity

The ability of mitochondria to sequester Ca^2+^ is an index of its Ca^2+^ loading capacity, without altering mitochondrial function. Here we calculated mitochondrial Ca^2+^ buffering capacity (mβ_Ca_) using the model described by Bazil et al. [42]. Briefly, experimental data for extra-and intra-matrix Ca^2+^ were fit with smooth trend curves satisfying the equation:(2)y(t)=p1+p2e(t−p3)p4+p5t,where y(t) was either [Ca^2+^]_e_ or [Ca^2+^]_m_ at any given time, t. Global trend-fits were performed in MATLAB (Mathworks, Inc., MA) and parameters p_1_ (offset value), p_2_ (pre = exponential constant), p_3_ (time lag), p_4_ (decay time constant), and p_5_ (steady-state slope) were estimated and optimized using the lsqnonlin and fmincon functions.

Mitochondrial Ca^2+^ buffering capacity for the second Ca^2+^ pulse (a cumulative of 40 µM added Ca^2+^) was then calculated [42] as:(3)$${m\beta }_{\mathrm{Ca}}=-\beta_{\mathrm{Ca},e}V_{r}\frac{d{\left[{\mathrm{Ca}}^{2+}\right]}_{e}}{\mathrm{dt}}/\frac{d{\left[{\mathrm{Ca}}^{2+}\right]}_{m}}{\mathrm{dt}},$$where, mβ_Ca_, is the intra-mitochondrial Ca^2+^ buffering power, β_Ca,e_ is the extra-mitochondrial Ca^2+^ buffering power determined by:(4)$$\beta_{\mathrm{Ca},e}=1+\frac{\partial {\left[\mathrm{CaEGTA}\right]}_{e}}{\partial {\left[{\mathrm{Ca}}^{2+}\right]}_{e}}.$$

Vr is the volume ratio of the extra-mitochondrial space and matrix space (~2000), d[Ca^2+^]_e_/dt and d[Ca^2+^]_m_/dt are the rates of change of extra-and intra-mitochondrial free [Ca^2+^], respectively. d[Ca^2+^]_e_/dt and d[Ca^2+^]_m_/dt were estimated by evaluating the analytical derivative of Equation (2) using parameter estimates obtained from the trend fits [42].

Trend fits for data in Appendix A were performed in Origin 2017 (OriginLab Corporation, Northampton, MA, USA).

### 2.8. Measurement of ΔΨ_m_, Redox State (NADH) and Matrix pH

Membrane potential was assessed by the dual-excitation ratiometric approach using the fluorescent dye, TMRM, as described by Scaduto and Grotyohann [40] and in our published work [35,38,39]. Fluorescence changes were determined by two excitations, λ_ex_ 546 and 573 nm, and a single emission λ_em_ 590 nm. The calculated ratio of λex 573/546 is proportional to ΔΨ_m_ and has the advantage of a broader dynamic range when compared to a single wavelength technique. Changes in mitochondrial redox state (NADH) were determined by autofluorescence (i.e., by exciting the energized mitochondria at λ_ex_ 350 nm and collecting data at λ_em_ 456 nm). An increase in the signal reflects an increase in the redox ratio of NADH to NAD^+^ (i.e., a shift to a more reduced state). Matrix pH was assessed using BCECF^AM^ (5 μM) at λ_ex_ 504 nm and λ_em_ 530 nm. This fluorescent probe emits less fluorescence in an acidic environment, thus a decrease in signal indicates matrix acidification and an increase in signal indicates matrix alkalization [29].

### 2.9. Depletion of Endogenous Mitochondrial Phosphate

Given the important role of P_i_ in the mitochondrial Ca^2+^ buffering system [14,29], we tested the effect of P_i_ in CsA-induced mitochondrial Ca^2+^ buffering. Isolated cardiac mitochondria were depleted of endogenous P_i_ by pre-incubating mitochondria for 10 min at room temperature with 0.75 units/mL hexokinase, 1 mM glucose, 0.5 mM ADP, 1 mM MgCl_2_, and 5 mM PA, as previously described [14,43,44].

### 2.10. Statistical Analyses

Data were transferred from PTI FelixGX (Version 3) into Microsoft^®^ Excel^®^ (2007). An unpaired Student’s t-test was used to evaluate significant differences between means of CsA-treated versus DMSO- and AdN-treated groups on specific variables ([Ca^2+^]_m_, [Ca^2+^]_e_, ΔΨ_m_, NADH, or pH_m_) in both Protocols A and B. The final data of a specific variable were expressed as mean ± standard error (SE) over at least 4 replicates of the same variable (*n* = 4). Comparisons within and between groups were performed by one-way ANOVA (analysis of variance) with Tukey’s post-hoc test to examine differences among individual groups. *p* < 0.05 (two-tailed) was considered significant. See Figure legends for statistical notations.

## 3. Results

### 3.1. Effect of CsA on Extra-Matrix Free [Ca^2+^]

To determine the effect of CsA on matrix Ca^2+^ uptake, we measured [Ca^2+^]_e_ during repetitive additions of 20 µM CaCl_2_ boluses at 5 min (300 s) intervals to allow characterization of the detailed kinetics of steady-state Ca^2+^ dynamics (influx and buffering). Figure 2 shows the dynamics of [Ca^2+^]_e_ during CaCl_2_ pulse challenges, with different treatments. Panels A, B, C and panels D, E, F depict the Ca^2+^ dynamics profile using Protocols A and B, respectively. Each panel consists of five traces representing different treatment groups (DMSO, ADP, OMN, OMN+ADP, and CsA) in the presence of approximately 40 μM EGTA (carried over from the isolation buffer). In response to each CaCl_2_ pulse, an increase in Fura-4F fluorescence intensity was observed, which then returned to a baseline; the steady-state (ss) level is marked by the flat response as mitochondria take up and sequester the added Ca^2+^. The opening of mPTP is evident by cessation of mitochondrial Ca^2+^ uptake and a sharp rise in the extra-matrix dye fluorescent intensity. Ca^2+^ concentrations were determined from the fluorescence ratios using Equation (1).

We plotted steady-state [Ca^2+^]_e_ (ss[Ca^2+^]_e_) as a function of cumulative added CaCl_2_ at each pulse (Figure 2B). The detailed dynamics of ss[Ca^2+^]_e_ for the initial four CaCl_2_ pulses, in each group, are illustrated in the enlarged scale inset (Figure 2A). The exposure of mitochondria to DMSO and OMN, followed by repeated boluses of CaCl_2_, resulted in a gradual increase in ss[Ca^2+^]_e_ with less mitochondrial Ca^2+^ uptake and rapid Ca^2+^ release by the third or fourth CaCl_2_ pulse. The total CRC for the DMSO and OMN were comparable (i.e., 133.3 ± 13.3 nmol Ca^2+^/mg protein and 146.6 ± 13.3 nmol Ca^2+^/mg protein, respectively) (Appendix A). In the presence of ADP, mitochondria took up more Ca^2+^ before pore opening and the CRC was further augmented with OMN+ADP; the CRC value increased from 213.3 ± 13.3 nmol Ca^2+^/mg protein for ADP alone to 373.3 ± 35.3 nmol Ca^2+^/mg protein for OMN+ADP (Appendix A). Mitochondria treated with CsA before the addition of CaCl_2_ boluses displayed a more robust Ca^2+^ uptake, with a significantly higher CRC value, 573.3 ± 26.6 nmol Ca^2+^/mg protein, compared with all other groups (Figure 2A and Appendix A). Importantly, in CsA-treated mitochondria, the addition of CaCl_2_ pulses (20 µM) did not significantly increase the ss[Ca^2+^]_e_ until the sixth to seventh pulse (Figure 2B), suggesting enhanced Ca^2+^ uptake. Figure 2C, summarizes the effects of the different treatments on the ss[Ca^2+^]_e_ for cumulative additions of 80, 140, and 180 µM of Ca^2+^, which corresponds to the fourth, seventh, and ninth CaCl_2_ pulses, respectively. The addition of CsA strongly blunted the Ca^2+^-induced increase in ss[Ca^2+^]_e_ by stimulating faster and more Ca^2+^ uptake. We observed that the ss[Ca^2+^]_e_ was significantly lower for the CsA-treated mitochondria than for OMN+ADP-treated mitochondria after the cumulative addition of CaCl_2_ of 80 µM (0.28 ± 0.0 µM vs. 0.50 ± 0.02 µM), 140 µM (0.57 ± 0.06 µM vs. 0.77 ± 0.06 µM), and 180 µM (0.71 ± 0.02 µM vs. 0.95 ± 0.1 µM) CaCl_2_, respectively (Figure 2B,C). The sustained low ss[Ca^2+^]_e_ for an extended period of CaCl_2_ additions in the CsA-treated group indicates a maintained ΔΨ_m_ for Ca^2+^ uptake, resulting in enhanced Ca^2+^ loading capacity because of improved buffering.

We next examined [Ca^2+^]_e_ dynamics in the situation in which mitochondrial matrix Ca^2+^ nearly reached threshold, as determined by the predicted opening of mPTP; in this case we added CsA just before the anticipated mPTP opening. Using Protocol B (Figure 1), [Ca^2+^]_e_ was measured and the kinetics were compared in response to adding either DMSO, ADP, OMN, OMN+ADP, or CsA just before the onset of the mPTP opening (Figure 2D). The dynamic changes in ss[Ca^2+^]_e_ during the addition of different treatments are illustrated in more detail in the inset of Figure 2D. In the DMSO-treated mitochondria, three to four Ca^2+^ pulses (cumulative addition of 68 ± 4.9 µM CaCl_2_) were sufficient to induce the release of matrix Ca^2+^. Addition of ADP or OMN reversed the initial pore opening and delayed matrix Ca^2+^ release by one to two pulses compared to DMSO. Addition of OMN+ADP also showed a significant reversal of Ca^2+^ release with reduction in the ss[Ca^2+^]_e_ (0.74 ± 0.18 µM). Thus, there was a considerable increase in the CRC by OMN+ADP compared to DMSO (Figure 2D and Appendix A) and a further delay in mPTP opening by one additional CaCl_2_ bolus compared to OMN or ADP alone. More impressively, the addition of CsA not only reversed the increasing trend of ss[Ca^2+^]_e_ to the baseline levels (0.42 ± 0.06 µM) (Figure 2D,E), it further maintained the ss[Ca^2+^]_e_ at a constant low value for an additional twelve to thirteen Ca^2+^ pulses. This resulted in a four-fold and a two-fold increase in the CRC, compared to DMSO and OMN+ADP, respectively (Figure 2E,F and Appendix A).

Altogether, these results demonstrate that CsA enhances mitochondrial Ca^2+^ uptake, thereby inhibiting a consequent increase in free [Ca^2+^]_e_ during CaCl_2_ pulse challenges, leading to an increase in the CRC of the mitochondria. This sustained low ss[Ca^2+^]_e_ and concomitant increase in Ca^2+^ uptake are likely explained by enhanced [Ca^2+^]_m_ buffering to maintain basal [Ca^2+^]_m_, which results in a preserved ΔΨ_m_ for Ca^2+^ uptake and greater CRC. To investigate further the potential for CsA on mediating Ca^2+^ buffering, it was necessary to examine the effects of CsA on matrix [Ca^2+^]_m_ dynamics in the next set of experiments.

### 3.2. Effect of CsA on Matrix Free [Ca^2+^] Handling

Matrix Ca^2+^ was assessed with Fura-4 AM as described in Materials and Methods. We explored the effect of CsA on [Ca^2+^]_m_, under identical conditions and protocols as shown in Figure 1 (Protocols A,B). Mitochondrial Ca^2+^ buffering was measured as a function of a decrease in Ca^2+^ fluorescence, reaching a steady-state at approximately 270 s after each bolus of CaCl_2_ added. The magnitude of mitochondrial Ca^2+^ uptake for the first CaCl_2_ pulse (20 µM) was similar in all groups; however, on subsequent additions of CaCl_2_, the ADP- and/or OMN-treated groups showed faster declines in [Ca^2+^]_m_ with lower mitochondrial steady-state [Ca^2+^]_m_ (ss[Ca^2+^]_m_) and delayed mPTP opening compared to DMSO (Figure 3A,B). Interestingly, the CsA-treated group showed a small increase in ss[Ca^2+^]_m_ with each CaCl_2_ pulse, but a gradual decline in ss[Ca^2+^]_m_ was observed after [Ca^2+^]_m_ exceeded 3 ± 0.10 µM with the cumulative addition of 100–150 µM CaCl_2_ (Figure 3A,B) and a significant increase in CRC up to fifteen to sixteen pulses. This suggested that the buffering effect of CsA on matrix Ca^2+^ is triggered when [Ca^2+^]_m_ reaches a certain value.

To estimate the mitochondrial Ca^2+^ buffering capacity, the ratio of bound Ca^2+^:free Ca^2+^ was calculated from the change in [Ca^2+^]_m_ (Figure 3A) to the total amount of Ca^2+^ taken up from the extra-matrix medium (ΣCa^2+^_uptake_): (ΣCa^2+^_uptake_-[Ca^2+^]_m_)/[Ca^2+^]_m_, as described previously [45]. Although the extent of bound Ca^2+^:free Ca^2+^ at each Ca^2+^ pulse was comparable in all the treated groups (Figure 3C), the addition of CsA maintained the buffering capacity, with a gradual increase in the capacity to bind Ca^2+^ up to fifteen or sixteen Ca^2+^ pulses (Figure 3C).

Greater uptake of Ca^2+^ from the extra-matrix space (indicated by lower ss[Ca^2+^]_e_), combined with lower ss[Ca^2+^]_m_, indicated a greater Ca^2+^ buffering capacity of mitochondria in the presence of CsA. Consistent with this notion, the calculated matrix Ca^2+^ buffering capacity (mβ_Ca_) in CsA-treated mitochondria was about ten-fold higher compared to DMSO and two-fold higher than with OMN+ADP (Figure 4). This CsA-mediated increase in mβ_Ca_ is possibly due to an effect of CsA in triggering the matrix physiological buffers to enhance sequestration of Ca^2+^.

After observing the high buffering capacity of mitochondria pre-treated with CsA before the CaCl_2_ bolus challenges, we next examined the effect of CsA on the rescue of mitochondria from Ca^2+^ release when the matrix Ca^2+^ buffering system (MCBS) becomes overwhelmed by the added boluses of CaCl_2_ (Figure 1B). As shown in Figure 3D,E, OMN and ADP, each failed to reverse the mitochondrial Ca^2+^ efflux with added boluses; however, adding CsA or OMN+ADP at similar time points significantly reduced ss[Ca^2+^]_m_ by reinstating Ca^2+^ sequestration. This reversal was more effective and sustained in the presence of CsA than with OMN+ADP. This observation is consistent with the calculated values of bound Ca^2+^: free Ca^2+^_,_ which increased two-fold for CsA compared to OMN+ADP (Figure 3F). Taken together, these data demonstrate that CsA increases the mitochondrial Ca^2+^ threshold for mPTP opening by activating [Ca^2+^]_m_ buffering that results in maintenance of a low ss[Ca^2+^]_m_.

### 3.3. Effect of CsA on Ca^2+^-Mediated Changes in ΔΨ_m_, NADH, and Matrix pH

A major driving force for Ca^2+^ uptake, in addition to the chemical gradient, is a high IMM potential gradient (ΔΨ_m_); but increased Ca^2+^ uptake without efflux or sequestration can decrease ΔΨ_m_ by flooding the matrix with positive charges. To strengthen the thesis that CsA increases the capacity of mitochondria to sequester Ca^2+^, we next investigated the effect of CsA on mitochondrial bioenergetics. ΔΨ_m_, NADH, and pH_m_ were assessed using the same protocols as described in Figure 1 for CRC to correlate changes in [Ca^2+^]_m_ to changes in bioenergetics over time. mPTP opening was marked by a sudden rise in the TMRM signal, indicating maximal depolarization of Ψ_m_. Correspondingly, the oxidation of NADH was marked by a decrease in matrix NADH signal intensity when mPTP opens. Figure 5 shows representative traces of ΔΨ_m_, NADH, and pH_m_ for each experimental condition. The rate of ΔΨ_m_ depolarization and NADH oxidation correlated well with the induction of mPTP, as seen in the CRC data. The loss of CRC coincided with total ΔΨ_m_ dissipation and NADH oxidation. DMSO-treated mitochondria (control) exhibited rapid Ca^2+^-induced ΔΨ_m_ depolarization and NADH oxidation (black trace) after only a few CaCl_2_ pulses. Addition of OMN+ADP significantly delayed the Ca^2+^ induced ΔΨ_m_ depolarization and NADH oxidation when compared to DMSO, with 533.3 ± 26.7 nmol Ca^2+^/mg protein vs. 200 ± 23 nmol Ca^2+^/mg protein and 546.7 ± 18.9 nmol Ca^2+^/mg protein vs.173.3 ± 13.3 nmol Ca^2+^/mg protein Ca^2+^ capacity, respectively (Figure 5A,B). Mitochondria treated with CsA maintained ΔΨ_m_ and NADH for a higher number of CaCl_2_ pulses than with OMN+ADP (666.7 ± 13.3 nmol Ca^2+^/mg protein, and 626.6 ± 13.3 nmol Ca^2+^ /mg protein, respectively) (Figure 5A,B). Mitochondrial matrix pH (pH_m_) is known to modulate mitochondrial P_i_ concentration and thus influence the matrix Ca^2+^ buffering [14]. The presence of CsA maintained pH_m_ at a basal level until mPTP opened (Figure 5C).

In Protocol B, intervention with OMN + ADP or CsA maintained ΔΨ_m_, mitochondrial NADH, and pH_m_ (Figure 5D–F), and contributed to the improved capacity of mitochondria to take up and sequester additional Ca^2+^ after CaCl_2_ pulses. However, OMN+ADP was less effective in preserving ΔΨ_m,_ NADH, and pH_m_ compared to CsA. This incapacity to sustain the bioenergetic status in the OMN+ADP- vs. CsA-treated mitochondria during CaCl_2_ challenges reflects a lower capacity to sequester Ca^2+^ in the matrix for a protracted time.

In summary, maintenance of ΔΨ_m_, NADH, and pH_m_ in the presence of CsA is consistent with changes in [Ca^2+^]_e_ and [Ca^2+^]_m_ that reflect greater Ca^2+^ sequestration (Appendix A) and uptake. Collectively, these results indicate that CsA reduced the accumulation of [Ca^2+^]_m_, by potentiating matrix Ca^2+^ buffering, which in turn, maintained ΔΨ_m_, NADH, and pH_m_ necessary for normal mitochondrial function. Together, these mitochondrial variables preserve mitochondria and protect against mPTP opening.

### 3.4. Time Dependent Effect of CsA Addition on Rescue of Mitochondria from Imminent Ca^2+^-Induced mPTP Opening

After demonstrating that CsA can reverse the induction of mPTP opening (Figure 2, Figure 3 and Figure 5), we next investigated the dynamics of [Ca^2+^]_e,_ [Ca^2+^]_m_ and ΔΨ_m_, by adding CsA at three different time points, before the onset of mPTP opening_._ This approach allowed us to determine the threshold at which CsA can effectively restore the mitochondrial sequestration system that will protect mitochondria from Ca^2+^ overload-mediated pore opening. Figure 6, panels A-C, show changes in [Ca^2+^]_e_, [Ca^2+^]_m_, and ΔΨ_m_ depolarization_,_ induced by adding CsA at 1, 2, and 3 min after the last CaCl_2_ bolus in which mitochondrial Ca^2+^ uptake was observed before pore opened. Right panels D-F show detailed (close up) comparison of kinetics of [Ca^2+^]_e_, [Ca^2+^]_m_, and ΔΨ_m_ after adding CsA at different time points. Adding CsA at all three tested time points, markedly delayed the large increase in [Ca^2+^]_e_ due to mitochondrial Ca^2+^ release. However, the effect of CsA to prolong Ca^2+^ uptake, which eventually maintains ss[Ca^2+^]_e_ at baseline, diminished as the interval before CsA addition and [Ca^2+^]_e_ accumulation was lengthened (Figure 6A). Adding CsA at 1 min caused a decline in [Ca^2+^]_e_, with a marked decrease in ss[Ca^2+^]_e_ (0.39 ± 0.07 µM) of the succeeding Ca^2+^ pulses, compared to adding CsA at 2 min (0.67 ± 0.03 µM) and 3 min (0.84 ± 0.05 µM) (Figure 6D; inset). In addition, we examined for changes in kinetics of [Ca^2+^]_m_ with CsA added at the same time points (Figure 6B). The rate of maximal Ca^2+^ buffering (i.e., the time to reach steady-state [Ca^2+^]_m_) and the Ca^2+^ threshold for pore opening was significantly higher when CsA was added at the early time points (i.e., 1 and 2 min) compared to the late time point of 3 min (Figure 6E, inset).

Next, in a parallel study, we monitored the corresponding changes in ΔΨ_m_ profile at the same rescue time points (1, 2, or 3 min). Adding CsA reversed the Ca^2+^-induced ΔΨ_m_ depolarization even after a large depolarization (i.e., at 3 min; Figure 6C,F). Similar to its effect on [Ca^2+^]_e_ and [Ca^2+^]_m_, CsA restored and maintained ΔΨ_m_ for a longer period at rescue points of 1 min vs. 2 and 3 min. Thus, at these points of intervention, CsA suppressed mPTP opening by increasing matrix Ca^2+^ buffering capacity, which maintained ΔΨ_m_ and the driving force for further Ca^2+^ uptake (Figure 6C). Together, these results demonstrate that the magnitude of CsA-mediated increase in Ca^2+^ threshold for mPTP opening and maintenance of mitochondrial integrity is dependent on the [Ca^2+^]_m_ level before CsA intervention.

### 3.5. Role of Inorganic Phosphate in CsA-Induced [Ca^2+^]_m_ Regulation.

Inorganic phosphate (P_i_) is a required component for mitochondrial matrix Ca^2+^ buffering [14,29]. To gain insight into the mechanism that underlies CsA-mediated activation of the MCBS, we monitored mitochondrial Ca^2+^ handling and ΔΨ_m_ during repeated boluses of 20 µM CaCl_2_ every 5 min, as described in Materials and Methods, but now in the absence of P_i_. With mitochondria depleted of P_i_, and in P_i_-free media, the CRC of mitochondria treated with CsA before the CaCl_2_ pulses was not different from DMSO (control). In addition, these mitochondria showed a gradual increase in ss[Ca^2+^]_e_ and interestingly, after cumulative additions of CaCl_2_ to 80 ± 15 µM, there was a significant decrease in mitochondrial Ca^2+^ uptake during additional CaCl_2_ pulses (Figure 7A). These results implicated a P_i_-dependent mechanism in the CsA-mediated delay in mPTP opening. In contrast, ADP and OMN+ADP, but not OMN alone, caused a significant delay in mPTP opening (Figure 7A) in the absence of P_i_.

Along with observing the P_i_-mediated effect of CsA on [Ca^2+^]_e_ dynamics, we also measured [Ca^2+^]_m_ under identical conditions. In the absence of P_i_, mitochondria showed a gradual increase in ss[Ca^2+^]_m_; matrix Ca^2+^ sequestration was strongly blunted in both DMSO-and CsA-treated groups. This reflected diminished buffering capacity with the increase in [Ca^2+^]_m_ (Figure 7B)_._ However, in the presence of ADP and OMN+ADP in the P_i_-depleted condition, mitochondria displayed robust CRC and enhanced Ca^2+^ buffering and thus decreased [Ca^2+^]_m_ (Figure 7B). Intriguingly, this effect was stronger than in the P_i_ replete condition (Figure 3). Mitochondria also showed an increased ratio of bound Ca^2+^:free Ca^2+^ in the OMN+ADP-treated group, but not in the DMSO and CsA groups (Figure 7D). These data further support the premise that P_i_ is crucial in CsA-induced matrix Ca^2+^ buffering and P_i_ is a requisite component of matrix calcium sequestration.

Since we observed significant attenuation of Ca^2+^ uptake and buffering by CsA in the absence of P_i_, we addressed how the altered mitochondrial Ca^2+^ dynamics impacted ΔΨ_m_. Analysis of ΔΨ_m_ in mitochondria depleted of P_i_ during CaCl_2_ bolus challenges revealed a gradual depolarization with each Ca^2+^ pulse over time in the DMSO-, OMN-, and CsA-treated groups (Figure 7C); this was consistent with the low CRC in these three groups due to the poor buffering after additional CaCl_2_ pulses. In contrast, mitochondria exposed to ADP or OMN+ADP in the P_i_-depleted state exhibited restored and sustained ΔΨ_m_, which supported a robust CRC (Figure 7C).

To further confirm the requisite role of P_i_ in mediating CsA-induced activation of the MCBS, a rescue experiment with 5 mM P_i_ was performed with DMSO-and CsA-treated groups in P_i_-depleted condition. With addition of deionized H_2_O (vehicle), pore opening was not prevented in either group (data not shown). The addition of exogenous P_i_ to the buffer triggered a rapid reversal of Ca^2+^ release (decrease in ([Ca^2+^]_e_) in parallel with complete restoration of ΔΨ_m_ (Figure 8). In contrast, additional Ca^2+^ pulses in the P_i_ free DMSO-treated group failed to maintain ss[Ca^2+^]_e_ and basal Ψ_m_, and induced rapid Ca^2+^ efflux (Figure 8). However, the CsA-treated mitochondria showed a robust uptake of [Ca^2+^]_e_ with low ss[Ca^2+^]_e_ and sustained ΔΨ_m_ maintenance with additional CaCl_2_ boluses (Figure 8). Taken together, these results establish that P_i_ is required for CsA-mediated mitochondrial Ca^2+^ buffering that maintains low [Ca^2+^]_m_ and preserves ΔΨ_m_; this in turn contributes to the capacity for more Ca^2+^ uptake and thus increases the Ca^2+^ threshold for mPTP opening.

## 4. Discussion

Matrix free [Ca^2+^] ([Ca^2+^]_m_) plays two important roles: (i) Activation of Ca^2+^-dependent dehydrogenases for oxidative phosphorylation at low concentrations [46]; and (ii) regulation of cytosolic Ca^2+^ by sequestration of excess Ca^2+^ at high concentrations [47]. Excessive accumulation of free [Ca^2+^]_m_ is a leading factor in inducing mPTP opening. It is well established that repetitive mitochondrial Ca^2+^ loading triggers a gradual increase in [Ca^2+^]_m_, leading to a loss of IMM integrity that results in dissipation of ΔΨ_m_ and release of Ca^2+^. CsA is known to delay pore opening, in part, by inhibiting the PPIase activity of Cyp-D [31]. Whether CsA-mediated delay in mPTP opening involves regulation of [Ca^2+^]_m_ by P_i_-induced matrix Ca^2+^ buffering has not been addressed before. In this study, we investigated the effects of CsA on [Ca^2+^]_m_ regulation during repeated Ca^2+^ loading and its functional significance in mPTP opening. Additionally, we determined if changes in [Ca^2+^]_m_ induced by CsA correlated with changes in mitochondrial bioenergetics under identical experimental conditions and if matrix P_i_ was required for the observed CsA effects.

Since the key postulate was that CsA contributes to mitochondrial Ca^2+^ buffering, all experiments were performed in Na^+^-free condition to completely block NCLX as a route for efflux of excess matrix Ca^2+^. This allowed us to directly assess mitochondrial Ca^2+^ buffering capacity under different treatments. Our major findings during repetitive CaCl_2_ bolus challenges are: (i) CsA maintained basal ss[Ca^2+^]_m_ owing to increased mitochondrial Ca^2+^ buffering capacity; (ii) the effectiveness of CsA to maintain basal ss[Ca^2+^]_m_ correlates well with preserved mitochondrial bioenergetics; (iii) the buffering effect of CsA in a P_i_-replete buffer was more pronounced than the known buffering effect of OMN+ADP; (iv) CsA-induced buffering was abolished in P_i_-depleted mitochondria and P_i_-free experimental medium. We conclude that the CsA-mediated delay in mPTP opening could, in large part, be attributed to CsA-induced activation of a P_i_-dependent mitochondrial Ca^2+^ buffering system (MCBS), which maintains a low free [Ca^2+^]_m_ and preserves mitochondrial bioenergetics.

### 4.1. CsA-Mediated Inhibition of mPTP Opening Relates to the ss[Ca^2+^]_m_

Using the two protocols (Figure 1A,B), we examined the changes in [Ca^2+^]_e_ and [Ca^2+^]_m_ in response to boluses of CaCl_2_ in the presence of vehicle (DMSO), CsA, ADP, OMN, or OMN+ADP over time. Our experimental approaches allowed us to define the contribution of CsA in the regulation of [Ca^2+^]_m_ when CsA was given before the CaCl_2_ boluses (Protocol A) and at the threshold for pore opening under condition of increased free [Ca^2+^]_m_ accumulation (Protocol B). Our results clearly indicate that the effect of CsA on delaying mPTP opening is due largely to its efficacy in maintaining free ss[Ca^2+^]_m_ by activating the MCBS in a P_i_-dependent manner, and thereby preclude early mitochondrial Ca^2+^ overload and delay induction of mPTP opening. Sustained low ss[Ca^2+^]_e_ in the CsA-treated group indicated increasing mitochondrial Ca^2+^ uptake driven by the enhanced sequestration of free [Ca^2+^]_m_ to maintain a transmembrane Ca^2+^ gradient and a charged ΔΨ_m_ that facilitated additional Ca^2+^ uptake (Figure 2). Unlike previous studies [14,33,34], NCLX was blocked under our experimental conditions, to prevent Ca^2+^ efflux during the repetitive CaCl_2_ additions; therefore, the net free ss[Ca^2+^]_m_ in our study was determined by the balance between Ca^2+^ uptake and Ca^2+^ sequestration.

Notably, the CsA-induced buffering of mitochondrial Ca^2+^ resulted in greater Ca^2+^ uptake to attain a steady-state, as shown by the gradual decrease in ss[Ca^2+^]_m_ with each added CaCl_2_ pulse (Figure 3). Insofar as Ca^2+^–P_i_ precipitation is a major mechanism for mitochondrial Ca^2+^ buffering, the sustained ss[Ca^2+^]_m_ after each CaCl_2_ bolus indicated matrix Ca^2+^ storage, likely in the form of various inorganic Ca–P_i_ complexes [14]. The low and maintained ss[Ca^2+^]_m_ during continuous matrix Ca^2+^ uptake is consistent with formation of these complexes. Although our study did not provide direct experimental evidence for CsA-induced matrix Ca–P_i_ complex formation, the continuous rise in estimated bound Ca^2+^:free Ca^2+^ ratio with each CaCl_2_ bolus as well as the ten-fold increase in mβ_Ca_ clearly reflects a CsA effect on [Ca^2+^]_m_ buffering capacity (Figure 3).

The protective effect of CsA in delaying mPTP opening has long been reported [28,31,32]. Our findings; however, provide the first direct evidence for a novel effect of CsA to enhance the capacity of mitochondria to sequester Ca^2+^ by which it obviates Ca^2+^-induced mPTP formation. Moreover, the effect of CsA in mediating greater matrix Ca^2+^ buffering explains the sustained free [Ca^2+^]_m_ reported by Chalmers and Nicholls [14] and the CsA-induced inhibition of mitochondrial Ca^2+^ efflux observed in other prior studies [33,34].

### 4.2. Underlying Mechanism of the CsA-Mediated [Ca^2+^]_m_ Regulation

It is well established that mitochondria are able to sequester large amounts of Ca^2+^, while maintaining free [Ca^2+^]_m_ over a range of 0.1 and 10 μM depending on the Ca^2+^ load [14]; however, the mechanism and kinetics for this are unclear. Matrix Ca^2+^ buffering capacity is determined by: i) The quantity of Ca^2+^ that can be retained, and ii) the Ca^2+^ threshold level for release when Ca^2+^ exchangers are blocked or maximally operated [48]. The role of P_i_ as a physiological buffer in regulation of [Ca^2+^]_m_ has been extensively studied [14,44,45,49]. The major mechanism of P_i_-mediated Ca^2+^ sequestration in mitochondria is believed to be achieved by formation of amorphous Ca^2+^–P_i_ complexes in the matrix [48,50,51], which in turn maintain the free [Ca^2+^]_m_ at a low level. Hence, sustained [Ca^2+^]_m_ cyclically promotes more Ca^2+^ uptake via the MCU due to better preservation of both the Ca^2+^ gradient and ΔΨ_m_.

Though P_i_ plays an essential role in matrix Ca^2+^ buffering, P_i_ has also been suggested to induce mPTP opening [52]. A recent study associated Ca^2+^–P_i_ precipitation with complex I inhibition and reduced ATP synthase rate during Ca^2+^ overload [53]. Another report demonstrated that increasing [P_i_] decreased the mitochondrial Ca^2+^ loading capacity [14]. It was suggested that the mPTP-sensitizing effects of P_i_ was likely due to its effect in decreasing matrix-free Mg^2+^, an mPTP inhibitor [20]. In addition, formation of polyphosphate, a known inducer of mPTP, could be a factor in regulating the Ca^2+^ threshold for mPTP activation [54,55]. Interestingly, two prior studies [56,57] indicated that P_i_ is necessary for the inhibitory effect of CsA on mPTP opening. However, two other studies reported that CsA inhibits mPTP opening even in the absence of P_i_ [58,59]. Conversely, in our study, the CsA-induced enhancement of matrix Ca^2+^ buffering was completely annulled when both mitochondria and the experimental medium were depleted of P_i_ (Figure 7). This loss of Ca^2+^ sequestration by CsA was reinstated when exogenous P_i_ was added just before activation of the mPTP (Figure 8). These observations provide the essential explanation for the requirement of P_i_ in the CsA-mediated MCBS and delay in mPTP opening.

The importance of mitochondrial matrix Ca^2+^ buffering via P_i_ is underscored by the studies of Wei et al. [44,45]. They reported that P_i_ modulates the total amount of Ca^2+^ uptake with smaller CaCl_2_ boluses, whereas P_i_ modulates Ca^2+^ buffering capacity with larger CaCl_2_ boluses. Since we had P_i_ in our experimental medium and the mitochondria were replete with exogenous P_i_, the observation that CsA induced low ss[Ca^2+^]_e_ and ss[Ca^2+^]_m_ could be explained by the following: (i) CsA activates P_i_-dependent matrix Ca^2+^ buffering potentially by maintaining the rate of Ca^2+^–P_i_ complex formation; and (ii) CsA may activate P_i_ transport processes (via H^+^/P_i_ transporter and/or phosphate carrier) that help to maintain both the IMM pH_m_ and ΔΨ_m_ gradients. These processes would limit the increase in free [Ca^2+^]_m_, which in turn would contribute to more Ca^2+^ uptake and retention by increasing the electrochemical driving force for Ca^2+^ influx.

### 4.3. CsA vs. ADP; As a Regulator of [Ca^2+^]_m_

AdN are implicated as one of the multiple matrix factors responsible for sequestering Ca^2+^ by mitochondria [29,60,61,62]. AdN can potentiate mitochondrial Ca^2+^ buffering by maintaining high matrix P_i_ concentrations that can facilitate precipitation of AdN-Ca–P_i_ complexes, including, ATP-Mg^2−^/P_i_^2−^ and HADP^2−^/P_i_^2−^, and thereby increase the Ca^2+^ threshold for mPTP opening [60,63]. In a study by Carafoli et al. [60], it was reported that mitochondrial Ca^2+^-buffering is proportional to mitochondrial ADP uptake. In our P_i_-replete study, OMN+ADP had a relatively small effect on Ca^2+^ buffering compared to CsA, but it had a significantly larger effect than ADP or OMN alone (Figure 2, Figure 3 and Figure 5). A reasonable explanation could be that OMN, an ATP synthase (Complex V) inhibitor [64], could contribute towards augmenting the AdN pool and thus enhance matrix Ca^2+^ buffering. Consistent with our findings, a previous study also showed a greater CRC with a low-concentration of ADP with OMN compared to 10-fold larger concentration of ADP alone [30]. Thus, in agreement with Sokolova et al. [30], the observed high buffering capacity and expanded CRC with OMN+ADP is largely attributed to the ADP component of the matrix AdN pool. However, a previous study [62] reported that AdN also prevent mitochondrial Ca^2+^ influx by directly chelating Ca^2+^ by a Ca–ATP complexation [61]. Contrary to this observation, in our study, the direct effect of ADP on binding free Ca^2+^ was negligible, as assessed by adding ADP and CaCl_2_ together in mitochondria-free experimental buffer (Appendix A). Additionally, carboxyatractyloside-mediated inhibition of ADP uptake via adenine nucleotide translocase precluded matrix Ca^2+^ buffering and blunted the CRC by OMN+ADP or ADP alone (Appendix A). In this case, the extra-matrix ADP that accumulated did not chelate the Ca^2+^ added to the buffer. Altogether, these observations indicate that a direct sequestration of Ca^2+^ outside the mitochondria does not explain the effect of ADP alone or OMN+ADP on the enhanced CRC in our study.

A previous study [42] from our group proposed that the MCBS relies on at least two classes of Ca^2+^ buffers. The first class could represent classical Ca^2+^ buffers, including mostly metabolites (ATP, ADP, and P_i_) and mobile proteins that bind a single Ca^2+^ ion at a single binding site. A second class of buffers could be associated with the formation of amorphous Ca^2+^ phosphates, which may be capable of binding multiple Ca^2+^ ions at a single site in a cooperative fashion [35,38,39,42]. Genge et al. [65] showed, in an in vitro study, that annexins, a diverse class of proteins, are required for Ca^2+^-phosphate nucleation. Additionally, many studies have suggested an AdN-dependent Ca^2+^-binding property of annexins [66]. Interestingly, mitochondria exposed to ADP alone or OMN+ADP retained their ability to maintain low ss[Ca^2+^]_e_ and ss[Ca^2+^]_m_ for an extended period of cumulative CaCl_2_ additions, and showed a higher Ca^2+^ threshold for mPTP opening without P_i_ compared to with P_i_ (Figure 7). This extended delay in mPTP opening in the P_i_-depleted state compared to the P_i_-replete state reflects the ability of P_i_ to induce early mPTP opening under certain conditions [52]. In this case, the presence of P_i_ appears to counteract the ADP delay effect and induce a much earlier pore opening compared to the P_i_-depleted state. The mechanisms for this AdN-mediated massive matrix Ca^2+^ loading capacity in the absence of exogenous P_i_ is unclear and needs to be further investigated. A plausible hypothesis could be that, in the absence of P_i_, a significant Ca^2+^ loading capacity of AdN might be mediated via direct interaction with annexins. CsA, on the other hand, might function as a mediator that activates a P_i_-dependent Ca^2+^ buffering system. Another possibility is that Cyp D, as a PPIase, reduces free phosphate levels in the matrix or blocks the Ca^2+^ binding property of annexins; this then would be relieved by CsA’s effect to block Cyp D.

### 4.4. Implication of CsA-Mediated Ca^2+^ Buffering on Mitochondrial Bioenergetics

Elevated [Ca^2+^]_m_ over the nanomolar range is reported to increase NADH generation in part by stimulating Ca^2+^-sensitive dehydrogenases of the TCA cycle [67,68] and activating the F_0_F_1_-ATP synthase [69], thereby accelerating oxidative phosphorylation (OXPHOS). However, excess mitochondrial free Ca^2+^ can dissipate ΔΨ_m_ and impede OXPHOS. The IMM ΔΨ_m_ is the key factor in generating the proton motive force across the IMM; it is also one of the primary driving forces for Ca^2+^ uptake via the MCU [70] and triggers Ca^2+^ efflux via the NCLX [71,72]. Therefore, if mitochondria continue to take up Ca^2+^ under increased extra-matrix Ca^2+^ exposure, the Ca^2+^ would have to be buffered or ejected to prevent excess free [Ca^2+^]_m_ accumulation that could dissipate ΔΨ_m_ and increase oxidation of NADH.

The stability of Ca^2+^–P_i_ precipitates inside the mitochondrial matrix largely depends on pH_m_ [50]. It is also proposed that the matrix [P_i_] depends on the pH gradient (e.g., a change in pH from 7 to 8 has been estimated to increase [P_i_] by a factor of 1000 [14,50]). Thus, matrix alkaline conditions could facilitate Ca^2+^–P_i_ precipitation, whereas matrix acidification could lead to a destabilization of the Ca^2+^–P_i_ precipitate and so enhance matrix free Ca^2+^ levels [14]. Consequently, we correlated the changes in [Ca^2+^]_m_ with indices of mitochondrial bioenergetics (ΔΨ_m_, NADH, and pH) (Figure 5) to have a better understanding of the CsA-mediated MCBS. Mitochondria exposed to CsA before the repetitive CaCl_2_ boluses, exhibited robust mitochondrial Ca^2+^ uptake and rapid [Ca^2+^]_m_ buffering while maintaining basal ΔΨ_m_, NADH, and an alkalinized pH_m_ until mPTP opened (Figure 5). Maintaining ΔΨ_m_ during excess Ca^2+^ uptake in the absence of functioning NCLX suggests a strong matrix buffering effect that is induced by CsA.

In Protocol B, when CsA was added just before the onset of pore opening, NADH and ΔΨ_m_ levels transiently increased but immediately returned to baseline with each added CaCl_2_ bolus. This transient depolarization and NADH oxidation with each addition of CaCl_2_ was not observed in Protocol A. The reason for this is unclear. Nonetheless, the observed transient oxidation of NADH helped to restore ΔΨ_m_ after Ca^2+^ induced transient depolarization before the next CaCl_2_ bolus (Figure 5D,E). The transient redox oxidation and ΔΨ_m_ depolarization suggest that the CsA added at the point just before mPTP opening activated MCBS more slowly compared to Protocol A. In addition, CsA maintained the pH_m_ gradient during prolonged Ca^2+^ pulse challenges (Figure 5C). This finding also likely excludes a contribution of the mitochondrial calcium–hydrogen exchange (mCHE) to the Ca^2+^ extrusion in the absence of NaCl. We have recently reported that CsA obviates mCHE activity at low extra-matrix pH [19]. However, based on our current results, it is likely that CsA triggered an enhancement of mitochondrial Ca^2+^ buffering so that the resulting low [Ca^2+^]_m_ and maintained ΔΨ_m_ and pH_m_ accounted for the inactivity of mCHE.

## 5. Conclusions

The salient observation of this study is that CsA mitigated mPTP opening by promoting the maintenance of a low [Ca^2+^]_m_, by stimulating and/or potentiating MCBS. Specifically, we showed that the presence of CsA, (i) significantly delayed the mPTP opening when compared to ADP or OMN+ADP (Protocol A); (ii) overturned the high amplitude increase in [Ca^2+^]_m_ (Protocol B); (iii) maintained pH_m_, redox state (NADH) and basal ΔΨ_m_, which maintains the driving force for more Ca^2+^ uptake and sequestration; and (iv) activates P_i_-dependent mitochondrial Ca^2+^ sequestration to delay mPTP opening.

Our study provides a novel insight into how CsA-mediates a delay in mPTP opening by activating the MCBS, which lowers ss[Ca^2+^]_m_ below the threshold for mPTP activation. This concept is shown in the scheme presented in Figure 9. Our finding supports the notion that CsA facilitates P_i_-dependent matrix Ca^2+^ buffering, which maintains matrix free Ca^2+^ and enables massive Ca^2+^ loading capacity, without diminishing the driving force for Ca^2+^ influx by maintaining ΔΨ_m_. CsA may delay mPTP opening by enhancing P_i_-dependent matrix Ca^2+^ buffering and by inhibiting Cyp D [31,32]. The culmination of these two mechanisms, and possibly others not yet identified, might be responsible for CsA protection against mitochondrial Ca^2+^ overload. Together, these findings add to our understanding of the mechanism of CsA-mediated modulation of mPTP. Importantly, we believe that therapeutic approaches targeted at regulating [Ca^2+^]_m_ homeostasis represent a promising strategy to reduce cardiac injury due to Ca^2+^ overload by delaying mPTP opening and preventing induction of apoptosis.

## Figures and Tables

**Figure 1 cells-08-01052-f001:**
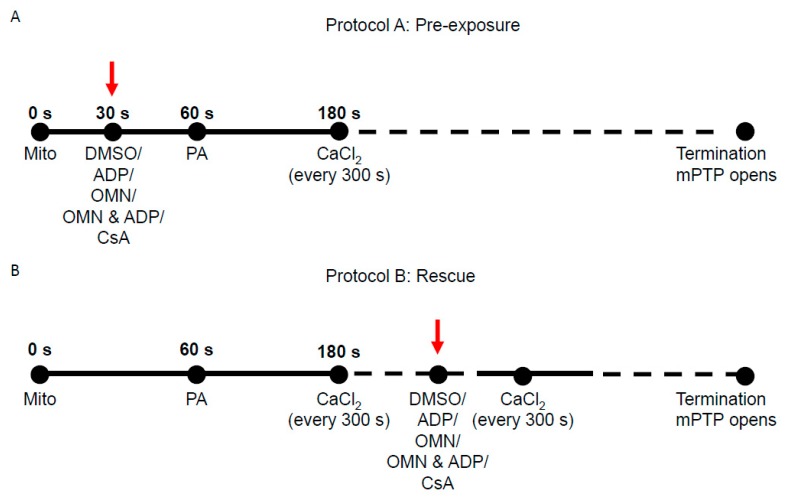
Schema of experimental timeline used to study the effect of Cyclosporin A (CsA) and adenine nucleotide (AdN) on mitochondrial Ca^2+^ handling and bioenergetics during repeated CaCl_2_ pulses. (**A**) In Protocol A, at t = 0 s, mitochondria (mito, 0.5 mg) were added to the Na^+^-free experimental buffer solution. The mitochondrial suspension was exposed to 0.5 μM CsA, 250 μM ADP, 10 μM oligomycin (OMN), or a combination of OMN+ADP at t = 30 s. Pyruvic acid (PA, 0.5 mM), was added at t = 60 s to energize mitochondria (state 2). At t = 180 s, 20 μM of CaCl_2_ was added, followed by sequential additions of 20 µM CaCl_2_ at every 300 s intervals until mPTP (mitochondrial permeability transition pore) opened or no further Ca^2+^ uptake was observed. (**B**) In Protocol B, the mitochondrial suspension was exposed to similar treatments as in Protocol A, but given after the last consecutive CaCl_2_ bolus preceding the imminent onset of mPTP opening.

**Figure 2 cells-08-01052-f002:**
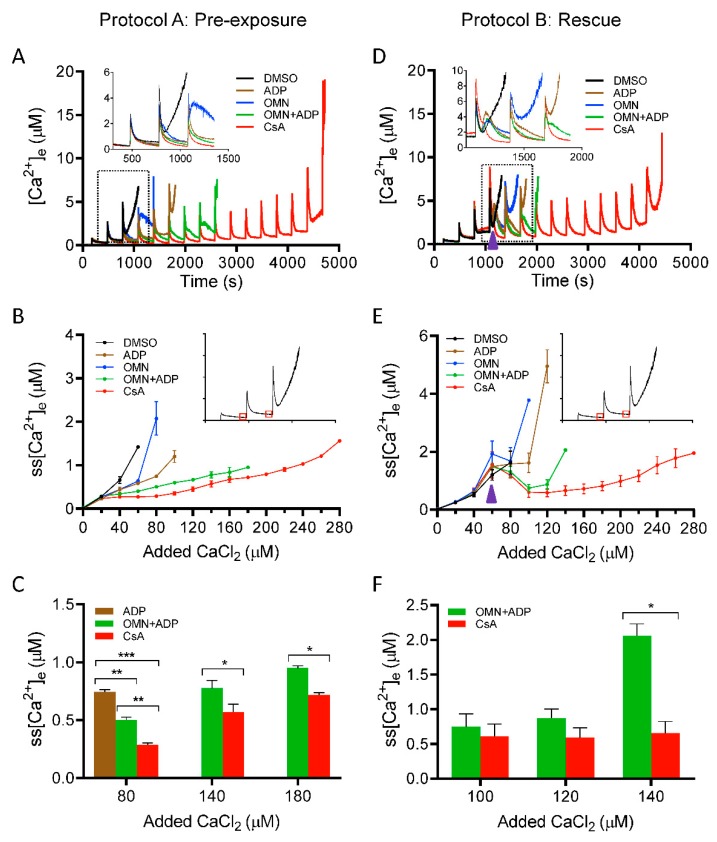
Effect of CsA and AdN on extra-mitochondrial calcium ([Ca^2+^]_e_) dynamics. Mitochondrial Ca^2+^ uptake and buffering for each of the treatment groups: DMSO (control; black trace), CsA (red trace), ADP (brown trace), OMN (blue trace), or OMN+ADP (green trace) are shown using the protocols depicted in Figure 1. Mitochondrial suspension was exposed to 0.5 μM CsA, 250 μM ADP, 10 μM OMN, or OMN+ADP before adding boluses of 20 μM CaCl_2_ (Protocol A; left column). Mitochondrial suspension was exposed to added boluses of CaCl_2_ (20 μM) and rescued mitochondria from mPTP opening (Protocol B; right column) with similar treatments as in Protocol A, at a time point at which it would initiate pore opening. Representative traces show change in extra-matrix free Ca^2+^ ([Ca^2+^]_e_) over time (**A**), and rescue of mitochondria from mPTP opening (**D**). Insets (**A**,**D**) show Ca^2+^ uptake kinetics in detail. Steady-state [Ca^2+^]_e_ (ss[Ca^2+^]_e_), 270 s after initiation of Ca^2+^ uptake, plotted as function of added Ca^2+^ (20 µM) every 300 s, in delay of mPTP opening (**B**), and rescue of mitochondria from mPTP opening (**E**). Insets (B,E) indicate the time points at which ss[Ca^2+^]_e_ was calculated. Quantification of steady-state [Ca^2+^]_e_ after a cumulative of 80, 140, and 180 µM CaCl_2_ during delay of pore opening (**C**) and cumulative of 100, 120, and 140 µM CaCl_2_ during rescue of mitochondria from mPTP opening (**F**). Error bars represent mean ± SEM (* *p* < 0.05; ** *p* < 0.01; *** *p* < 0.005). Arrowhead indicates time of addition of DMSO, ADP, OMN, OMN+ADP, or CsA during Protocol B.

**Figure 3 cells-08-01052-f003:**
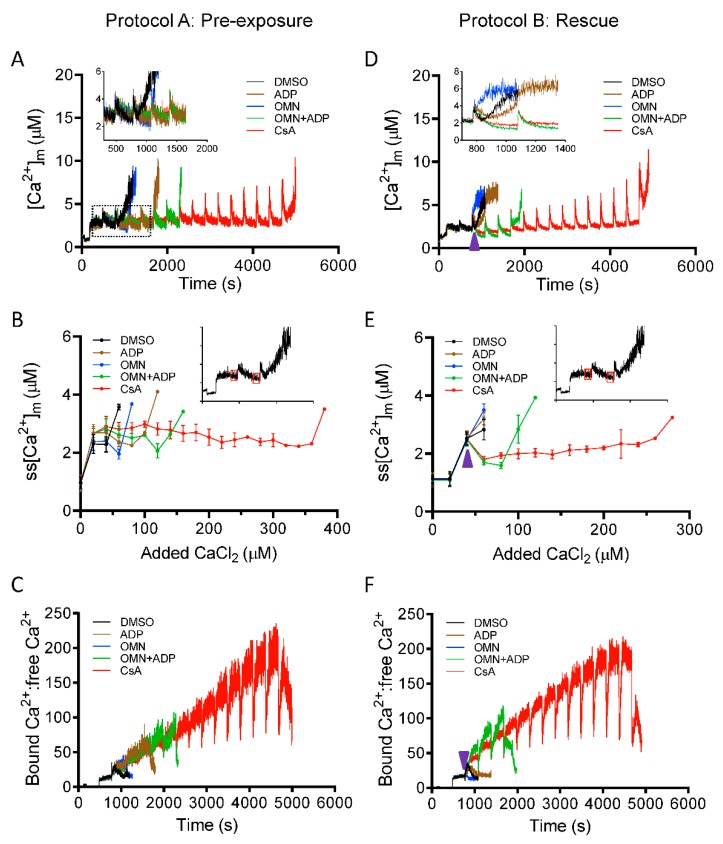
Effect of CsA and AdN on intra-matrix free Ca^2+^ ([Ca^2+^]_m_) dynamics. Mitochondrial Ca^2+^ uptake and buffering for each treatment groups, DMSO (control; black trace), CsA (red trace), ADP (brown trace), oligomycin (OMN, blue trace), or combination of OMN+ADP (green trace) are shown using the protocols depicted in Figure 1. Mitochondrial suspension was exposed to 0.5 μM CsA, 250 μM ADP, 10 μM OMN, or OMN+ADP before adding boluses of 20 μM CaCl_2_ (Protocol A; left column). Mitochondrial suspension was exposed to added boluses of CaCl_2_ (20 μM) and rescued from mPTP opening (Protocol B; right column) with similar interventions as in Protocol A, at a time point at which it would initiate mPTP opening. Representative traces show changes in [Ca^2+^]_m_ over time in delay of mPTP opening (**A**) and rescue of mitochondria from mPTP opening (**D**). Insets (A,D) show Ca^2+^ uptake kinetics in detail. Steady-state [Ca^2+^]_m_ (ss[Ca^2+^]_m_), 270 s after initiation of Ca^2+^ uptake, plotted as function of added Ca^2+^ (20 µM) every 300 s in delay of mPTP opening (**B**) and rescue of mitochondria from mPTP from opening (**E**). Insets (B,E) indicate the time points at which ss[Ca^2+^]_m_ was calculated. Change in matrix-bound Ca^2+^:free Ca^2+^ over time in delay of mPTP opening (**C**) and rescue of mitochondria from mPTP opening (**F**). Arrowhead indicates time of addition of DMSO, ADP, OMN, OMN+ADP, or CsA during Protocol B.

**Figure 4 cells-08-01052-f004:**
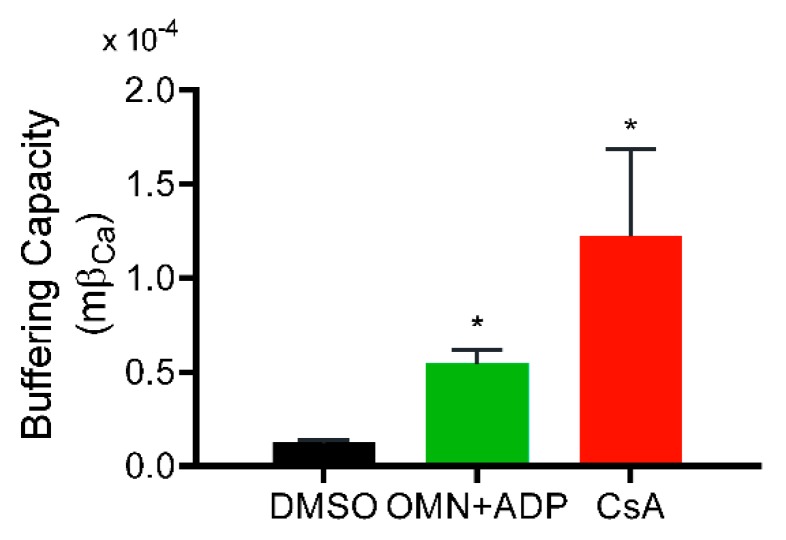
Effect of CsA and AdN on mitochondrial Ca^2+^ buffering capacity. Mitochondrial Ca^2+^ buffering capacity calculated from trend fits of [Ca^2+^]_e_ and [Ca^2+^]_m_ for DMSO-(control), CsA-, and OMN+ADP-treated mitochondria as described by Equations (2)–(4) in Materials and Methods. Buffering capacity for each treatment was calculated from three-five experiments each for [Ca^2+^]_e_ and [Ca^2+^]_m_ and averaged. Error bars represent mean ± SEM (* *p* < 0.01 compared with DMSO).

**Figure 5 cells-08-01052-f005:**
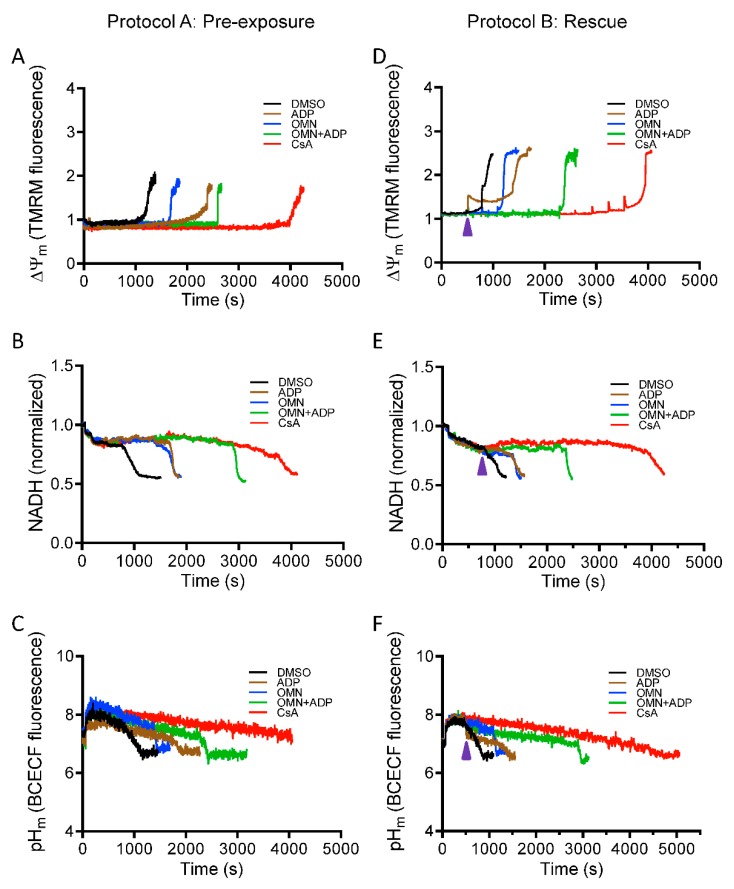
Effect of CsA and AdN on mitochondrial bioenergetics. The bioenergetic responses during Protocols A (left column) and B (right column) were monitored using the ΔΨ_m_ sensitive dye TMRM (tetramethylrhodamine methyl ester perchlorate) (**A**,**D**), NADH autofluorescence (**B**,**E**), and pH_m_-sensitive dye BCECF^AM^ (2′,7′-Bis-(2-Carboxyethyl)-5-(and-6)-carboxyfluorescein, acetoxymethyl ester AM) (**C**,**F**). Purple arrowhead indicates time of addition of DMSO (1 µM), ADP (250 µM), OMN (10 µM), OMN+ADP, or CsA (0.5 µM) during Protocol B.

**Figure 6 cells-08-01052-f006:**
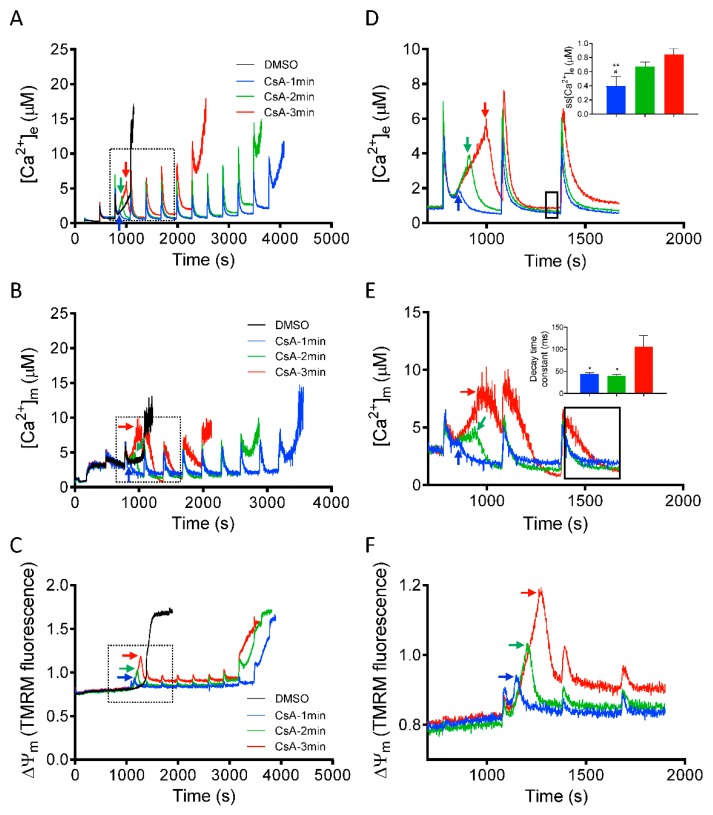
Time-dependent effects of CsA on mitochondrial Ca^2+^ dynamics and bioenergetics during rescue of mitochondria from mPTP opening. Changes in (**A**) extra-matrix free Ca^2+^ ([Ca^2+^]_e_), (**B**) intra-matrix free Ca^2+^ ([Ca^2+^]_m_) and (**C**) ΔΨ_m_, when CsA was added at 1 min (blue trace), 2 min (green trace), and 3 min (red trace) after the last Ca^2+^ bolus before another Ca^2+^ bolus would have caused mPTP opening. Right panels show the effect of CsA on (**D**) [Ca^2+^]_e_, (**E**) [Ca^2+^]_m_, and (**F**) ΔΨ_m_ dynamics during rescue of mitochondria from mPTP opening in greater detail. Insets (**D**,**E**) show relative ss[Ca^2+^]_e_ and decay time constant (ms) at specified time points (black dotted box), respectively. Arrows indicate time of addition of CsA (0.5 µM). Error bars represent mean ± SEM (* *p* < 0.05, ** *p* < 0.01 vs. 3 min and ^#^
*p* < 0.05 vs. 2 min).

**Figure 7 cells-08-01052-f007:**
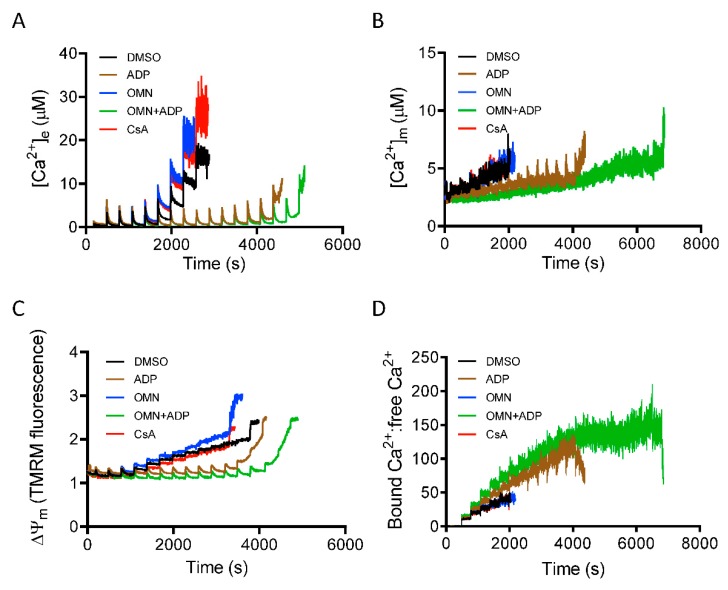
Effect of P_i_ on CsA-induced mitochondrial Ca^2+^ handling and bioenergetics. Time course of [Ca^2+^]_e_ (**A**), [Ca^2+^]_m_ (**B**), ΔΨ_m_ (**C**), and matrix-bound Ca^2+^:free Ca^2+^ (**D**) during consecutive additions of 20 μM CaCl_2_ to a suspension of P_i_-depleted mitochondria, pre-exposed to DMSO (control), CsA, ADP, OMN, or OMN+ADP.

**Figure 8 cells-08-01052-f008:**
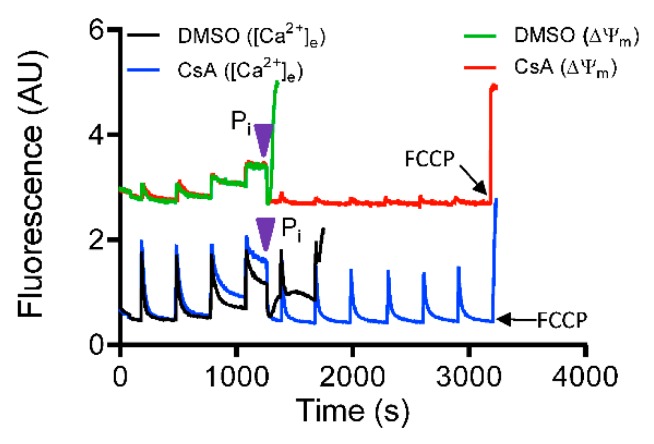
Mitochondrial Ca^2+^ modulation by CsA is phosphate (P_i_)-dependent. Representative traces show change in extra-matrix Ca^2+^ fluorescence (Fura-4F Ratio) and ΔΨ_m_ during consecutive 20 μM CaCl_2_ boluses to induce mPTP opening in P_i_-depleted mitochondria. P_i_ was added (purple arrowhead) at threshold point when mitochondria exhibited limited uptake of Ca^2+^ from the buffer.

**Figure 9 cells-08-01052-f009:**
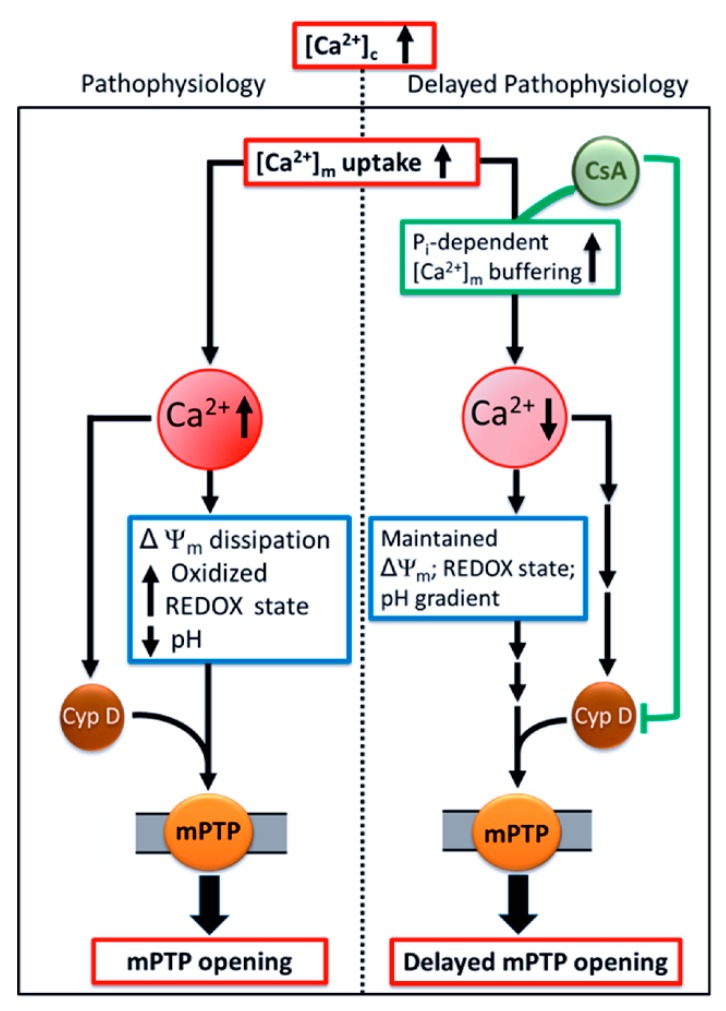
Schema of the potential mechanism by which CsA mediates delay in Ca^2+^-induced mPTP opening. Pathological conditions, like cardiac ischemia-reperfusion injury, leads to an increase in cytosolic Ca^2+^ ([Ca^2+^]_c_). This in turn increases [Ca^2+^]_m_ and generation of reactive oxygen species (ROS), impairs respiration and substrate utilization, and leads to uncoupling of oxidative phosphorylation. Lower ΔΨ_m,_ oxidized redox state, and dissipation of the pH_m_ gradient, together induces mPTP opening which triggers apoptosis. These detrimental consequences that underlie IR injury could be mollified by CsA, which allows the mitochondria to maintain their basal [Ca^2+^]_m_ via enhanced P_i_-dependent matrix Ca^2+^ buffering, in addition to, or through, Cyp D inhibition. Sustained low [Ca^2+^]_m_ maintains mitochondrial integrity and function and delays mPTP opening.

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
