# Peer review of "Cyclosporin A Increases Mitochondrial Buffering of Calcium: An Additional Mechanism in Delaying Mitochondrial Permeability Transition Pore Opening"

_cells, 2019, doi:10.3390/cells8091052_

Round 1
Reviewer 1 Report
In this study, Mishra and colleagues address the ongoing controversy between groups that propose that phosphate is required for the protective effects of cyclosporin A (CsA) on permeability transition (PT) versus groups proposing the opposite. The authors elegantly show in this study that CsA is no longer efficient as a PT inhibitor when isolated mitochondria lack phosphate in the incubation media and further propose that the protective effects of these inhibitors are partly related to the activation of a -yet to be identified- calcium buffering system within the mitochondrial matrix. While the scientific design of the study is sound and the take-home message is clear, this reviewer would like the authors to address some concerns before further considering this manuscript.
1) What is the possibility that the activation of the CsA-Pi-induced MCBS monitored by the authors is a secondary effect due to primary PT inhibition and enhanced mitochondrial coupling?
2) Have the authors tested lower phosphate concentrations i.e. 1mM?
3) The authors should consider the work by Kushnareva et al. (PMID 10049510) and Malyala in the Introduction and Discussion sections.
4) How can the authors be sure that mitochondria were depleted of endogenous Pi by preincubating with hexokinase, glucose, ADP and MgCl2 for 10 min in their mitochondrial samples?
5) It is somewhat difficult to make comparisons between ADP + Oligo versus CsA considering the very different targets of the inhibitors tested. What was the rationale behind doing such comparative study.
6) The authors are using high levels of oligomycin (10uM). Please note this may have unspecific effects. Please mention the type and source of oligomycin.
7) Have the authors tried using oligomycin + CsA in order to test its effects in a setting where ATP synthase is inactive even for endogenous AdNs?
8) Are the effects elicited by CsA recapitulated by CypD deletion? Perhaps this experiment would unequivocally show whether CypD is actually involved in the elusive MCBS regulation.
9) The abstract could be shortened by omiting the methods used.
10) What happens to matrix pH under phosphate depletion conditions?
Reviewer 2 Report
In their manuscript the authors evaluated the role of cyclosporine A in increasing the calcium buffering capacity of cardiac mitochondria. With respect to the available papers on this matter, the analyses have been performed over extended time period and the indispensable role of inorganic phosphate (Pi) has been established. The study is well conducted, based on solid mathematical modeling and the results support the authors’ conclusions. I’ve only few considerations.
-Previous papers have addressed a topic very similar to the one addressed in the present manuscript (see ref 33 and 34). To reinforce the originality of their work the authors should better stress the novel insight that their results bring on the protective role of CycA against mitochondrial calcium overload and permeability pore opening. Also, the role of Pi has long been debated, in contrast to the present manuscript and others, several papers consider Pi a pore opener. A deeper discussion of the results of the authors in such a context should help bringing light on the controversy.
-Isolated mitochondria and fluorescence spectrometry were used to measure calcium concentrations within the mitochondrial matrix or in the extra-mitochondrial space. Although a brief hint has been given in lines 174-176, the authors should better describe how were intra- and extra- mitochondrial fluorescent signals distinguished. Did the authors performed blank experiments without putting mitochondria in the recording solution to exclude experimental artifacts?
-The authors state that in protocol B the calcium kinetic in the presence of the various compounds was analyzed just before the onset of the mPTP opening. The authors should better explain how was that timing predicted and which parameter they used to do so.
-Lines 84-99 of the introduction are dedicated to a description of the results that is too detailed to be inserted in this section. As an appealing anticipation of the results, lines 99-101 are enough.
-Lines 179: BCECF, please describe the acronym in plain when mentioning for the first time.
-Line 218: the first figure is indicated as figure 4.
-Figure 2: the triple asterisk appears in the figure (panel C) but not in the figure legend, please amend.
Reviewer 3 Report
This manuscript deals with very important subject of the relationship between total calcium, free calcium and activation of the mPTP. The key finding of the work is that CSA dramatically increases calcium buffering capacity and delay PT activation. The overall experimental design is good and results support the conclusion. The key issue with this study is that it largely reproduces previous work by Chalmers and Nicholls. It is sited expensively (ref 14) throughout the text in the context of Ca-Pi granules but in fact that work specifically deals with bound/free calcium relationship and shows clearly increase in buffering capacity but not in free calcium in the presence of CSA. Further, the same paper by Chalmers reports TMRM and NAD(P)H assays as well. Ironically Ref. 14 doesn’t have any granule measurements (J Neurochem. 2007 Aug;102(4):1346-56 would have been the appropriate citation for DG imaging as a function of Ca load). Altogether my recommendation would be for authors to carefully review Chalmers and Nicholls work and clearly establish what their study adds to that and what is the novelty. I believe it is totally ok to reproduce and confirm previous studies as long as this is clearly outlined. In the revised version I encourage authors to carefully read and discuss the mini review that followed the Chamers paper “The Integration of Mitochondrial Calcium Transport and Storage” JBB 2004 in which they carefully discuss the relationship between Ca and Pi.
Round 2
Reviewer 1 Report
The concerns raised by this reviewer have been satisfactorily addressed. Therefore, I have no more comments for the Authors.
Reviewer 2 Report
The authors have satisfactorily answered to all my concerns, I have no more questions.
Reviewer 3 Report
In this referees assessment authors arguments confirm that current work weakness is originality. Specifically:
Our data agree with those of Chalmer’s et al. In fact, our study reproduces the observed CsA effect on constant free matrix Ca2+ over a prolonged period of Ca2+ loading.
Yes, agree with this statement.
"However, unlike Chalmer’s and Nicholl’s 2003 study, that has Na+ in the experimental buffer with potential NCLX activation, the NCLX was not active under our experimental conditions (see Material & Methods). Therefore, our experimental conditions completely ruled out mitochondrial Ca2+ efflux as ways to regulate matrix free Ca2+, and provides strong evidence of enhanced Ca2+ sequestration, as the only means for CsA-mediated delay in mPTP opening. This notion was speculated in the review by Chalmer’s and Nicholl’s, 2004."
In Chalmers Nicholls - effllux effects were negliegable - this can be easily determined by the very small levels of extramitochondiral calcium that was measured following bolus additions.
"In addition, using the Pi depleted mitochondria and Pi-free experimental condition, our study provides the underlying mechanism for CsA-mediated enhanced matrix Ca2+ buffering which maintains constant matrix free Ca2+ and enables massive Ca2+ loading capacity."
They did do that ...Fig 5 of the JBC paper
"Lastly, in contrast to continuous infusion (used by Chalmer’s and Nicholl’s 2003) our experimental approach allowed us to study the effect of bolus addition of Ca2+ on ΔΨm, NADH and pHm in relation to [Ca2+]m regulation in the presence of CsA."
They did do that to start ... this is Fig 1 of the JBC paper bolus vs continuous and then decided to go for continuous.
Round 3
Reviewer 3 Report
Yes , OK to publish.